# TCD: TEXT IMAGE CHANGE DETECTION FOR MULTI-LINGUAL DOCUMENT COMPARISON

## ABSTRACT

In general, the core technology used in imaged document comparison is based on Optical Character Recognition (OCR). However, the main drawbacks of using OCR for document comparison are that most users have to pick relevant language models for each document. Moreover, a multilingual document needs a multilingual OCR model, or a hybrid model has poor recognition performance. To overcome such drawbacks, we propose common Text image Change Detection (TCD) model for multilingual documents that utilize the unit-level text image-to-image comparison instead of text recognition. Our model generates the change segmentation maps in both directions from source to target and target to source. Furthermore, we propose to use the correlation between multi-scale attention features, which mitigates pre-processing of text image position and scale alignment. We created test data from printed and scanned documents in different languages and added public datasets such as Distorted document images (DDI-100), and Document binarization dataset (LRDE DBD). Finally, we compare the performance of our model with state-of-the-art semantic segmentation and change detection (CD) models, and also with OCR models. Experimental results on benchmarks demonstrate that our model outperforms other semantic segmentation models relatively by a good margin and meets the similar performance that of OCR methods.

## 1 INTRODUCTION

With the development of computer vision (CV), numerous documents in analog format have been transformed into a digital format that is computer-recognizable. Due to this evolution, various techniques to digitize analog documents such as paper documents have been suggested. Early OCR methods used pattern matching by comparing standard character templates for character recognition (TAUSCHEK, 1935; Schantz, 1982; Mori et al., 1992). Recently, with Deep Learning (DL) advancements (Krizhevsky et al., 2017; Simonyan & Zisserman, 2014; Girshick et al., 2014), DL techniques have been applied to various CV applications such as image classification, detection, and segmentation for improving performance. Image recognition got big evolution with an application of DL techniques, and at the same time. OCR also has shown dramatic improvement while various research with DL (Breuel et al., 2013; Anil et al., 2015; Lee & Osindero, 2016).

OCR has been used in the wide field including invoice, banking, legal, and digitization of paper documents because of their performance improvement (Stoliński & Bieniecki, 2011; Singh et al., 2012). Some tasks can be performed using OCR only, however, the technologies related to documents such as scanned documents comparison (Andreeva et al., 2020), or forgery detection (Ahmed & Shafait, 2014; Sirajudeen & Anitha, 2020) is consist of various techniques like layout analysis, or document de-skewing (Cai et al., 2021; Xu et al., 2020). Recently, with the advent of Transformer (Vaswani et al., 2017), various document analysis techniques using transformer are still in progress.

With vigorous research in document analysis (Huang et al., 2022; Fang et al., 2021), this served as a momentum for document comparison technology to be released as off-the-shelf software in the field of research. Most of the released off-the-shelf imaged document comparison software are based on OCR (Tafti et al., 2016). Most document comparison technologies combine processes such as pre-processing, structural analysis, text detection, and various post-processing for the performance of their core technology, OCR. In the end, when it comes to comparing OCR-based documents, the key is the performance of the OCR. It looks like a good performance in typically, but recognition

often fails for a variety of other reasons, including background noise, or some characters similar to numbers In particular, as the number of languages to be recognized increases, it is very difficult to maintain OCR performance consistently.

Meanwhile, the development of technologies such as the internet, logistics, and transportation have made society more global, resulting in more international transactions and contracts (Mukherjee, 2008; Bookbinder & Matuk, 2009). If the languages used by the contracting parties are different, the contract is generally written using both languages. Recognizing more than one language is a burden for OCR in itself. In particular, OCR-based document comparison technology shows limitations in situations such as recognizing languages with no information given in advance or untrained languages.

We overcome this limitation using image change detection without recognition. Our model compares directly between text area images instead of text recognition. Each text image comparison checks the differences in targets based on the source text area and also in opposite direction, it makes to know the difference between source for target and target for source.

We can summarize our contribution as follows:

- To our best knowledge, we first propose a text image-based two-way change detection model, which is a method independent of the text language.
- We use a correlation marginalization process based on the surroundings feature, so our model doesn't need any pre-processing.
- We present a new text image change detection test dataset where our algorithm shows the state-of-the-art performance and the component-wise effectiveness is analyzed through the ablation studies.

## 2 RELATED WORK

Text comparison is usually split into two subtasks: text detection and text segmentation. In our proposed method, we use semantic segmentation to perform image-to-image comparisons instead of text recognition. In this way, in document comparison, the core technology is the ability to detect the difference between two documents, such as recognizing or comparing two text images. In practice, document comparison is often preceded or followed by a variety of technologies, such as noise removal and document layout analysis, that are added to improve the performance of the core technology. We give a brief overview of relevant works below.

### 2.1 TEXT DETECTION

Text detection is performed as a first step in the series of processes that understand contents like text in images. The early approach was a heuristic way of using computer vision such as connected component or sliding window (Epshtein et al., 2010; Lee et al., 2011). Since then, the success of early deep learning models (Krizhevsky et al., 2017), and the proven effectiveness of convolutional neural networks (CNN) for object detection (Redmon et al., 2016; Ren et al., 2015), more recent approaches utilize detection model in text detection (Huang et al., 2019; Zhong et al., 2019). In the context of scanning or capturing documents, Jung et al. (2021) utilized a modified 2D Gaussian score map with a rectangular shape to make it robust to noise in scanned documents. Considering the various text shapes, Wang et al. (2019) proposed a network that separates overlapping text regions by applying pixel aggregation, which borrows the concept of clusters, and Zhu et al. (2021) introduced to represent text contours in the frequency domain, especially for highly-curved shapes. Liao et al. (2020; 2022) proposed a method to integrate the binarization process, one of the important post-processing in segmentation, into the segmentation network to effectively binarize text regions for real-time text detection.

### 2.2 TEXT RECOGNITION

Once the text has been detected through the previous steps, the text area is cropped and fed into the recognition model in order to recognize the letters at the position. Text recognition has also been studied for a long time, paired with text detection (Lin et al., 2020). Similar to text detection,

early text recognition approaches also have used CV and early machine learning techniques (Sarfraz et al., 2003; Smith, 2007), But recent research has mostly used neural networks. Graves et al. (2006) proposed Connectionist temporal classification (CTC) that decodes features using recurrent neural networks (RNN), This is particularly useful for problems where the length of the input sequence can vary. To borrow this idea, Yao et al. (2014); Shi et al. (2016) propose to use CNN and RNN to encode the sequence features using CTC for character alignment. Plus, Li et al. (2022) proposed a lightweight model using CTC and attention. On the other hand, As an approach to language-independent recognition, Huang et al. (2021) proposed that an applied Language Prediction Network (LPN), can recognize all languages with one weight.

### 2.3 SEMANTIC SEGMENTATION

Semantic segmentation is a computer vision technique that involves dividing an image into multiple regions, where each segment corresponds to a specific object or part of the image. It has applications in many fields, including self-driving cars, medical imaging, and augmented reality. Deep learning techniques, such as CNN, also have been used extensively for semantic segmentation tasks in recent years. One of the early deep learning models for semantic segmentation is the Fully convolutional network (FCN) (Long et al., 2015). Another popular architecture for semantic segmentation is the U-Net (Ronneberger et al., 2015). U-Net is an encoder-decoder architecture that uses skip connections to fuse features from different resolution levels. More recently, He et al. (2017) and Chen et al. (2018) have shown state-of-the-art results in many benchmarks. Mask R-CNN extends Faster R-CNN (Ren et al., 2015) object detection framework by adding a branch for predicting a binary mask for each detected object. Following the great success of transformers (Vaswani et al., 2017), Xie et al. (2021) proposed Segformer a simple and efficient segmentation model using transformers. Segmentation is also frequently used as change detection, It can segment images into pixels either changed or not. There are various approaches to change detection, recently, the transformer-based approaches also show the best performance such as Chen et al. (2021; 2023).

## 3 PROPOSED METHOD

Document text image change detection aims to detect changed text locations such as changed, added, or deleted. In a typical document comparison process, for finding some changed area, document comparison extracts texts from both source and target document images, recognizes them, and compares each other. Our proposed method merges recognition and comparison into text unit change detection. Overall, our proposed model aims to compare text area image pair from source and target document is same or different at a character level. We design our problem as a 2-way semantic segmentation model, from source to target and vice versa. In this section, we present the overall architecture of our model and key modules such as the marginalized multi-scale feature correlation map.

### 3.1 OVERALL ARCHITECTURE

Our proposed architecture is shown in Figure 1, it is based on Encoder-Decoder (Badrinarayanan et al., 2017). We aim to compare the two input images, so we feed them into an Encoder that shares the same weights. It is similar to Siamese network (Chopra et al., 2005; Koch et al., 2015) architecture. We use Resnet (He et al., 2016) as a backbone in the Encoder module. Text unit images contain small characters such as subscripts, punctuation, etc. So we extract multi-level features from the backbone, and with Lin et al. (2017) for preserving small characters that are easily ignored by the conv layer. In the image comparison, it is important to match the scale or alignment between images. However, in the Encoder, we use cross and cross-self attention to enhance image features and focus meaningful locations in feature space. In addition, we also use the correlation between feature maps, so our model doesn't need any pre-processing such as alignment.

After encoding the source and target images, each feature is fed to Decoder. Decoder generates the change segmentation map from each encoded feature. Unlike Encoders, Decoder has two parallel modules for each one of the images. The model consists only of Conv layers without fully connected blocks, so images of various sizes can be used. Decoder also uses features from the backbone and enhances lower resolution segmentation map using cross and cross-self attention.

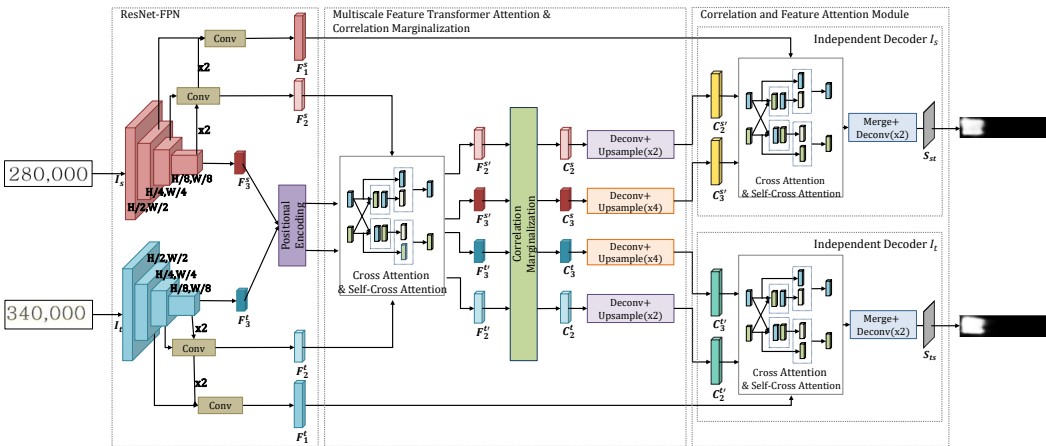

Figure 1: Proposed Text image Change Detection Model Architecture.

## 3.2 BACKBONE NETWORK

Given pair of input images $I_s$ and $I_t$ of size $(H, W)$, our Resnet with FPN network produces 3 multi-scale feature pyramid map pairs of sizes $(H/2, W/2)$, $(H/4, W/4)$, and $(H/8, W/8)$ for each image named as $F_1^s, F_2^s, F_3^s$ and $F_1^t, F_2^t, F_3^t$. Our backbone is based on the network proposed in Cheng et al. (2017) which is based on Resnet, and we use only up to the 3rd bottleneck layer. ResNet backbone output is passed through the FPN to improve the feature maps further. The FPN network has 3 multi-scale feature maps with output channel sizes are 64,64 and 512 respectively.

## 3.3 POSITIONAL ENCODING AND MULTI-SCALE FEATURE MAP ATTENTION

Taking inspiration from Dosovitskiy et al. (2020), We use the 2D extension of the standard positional encoding in Transformers following Carion et al. (2020) and Sun et al. (2021). We only add them to the FPN output feature map $F_3$ only because of the small spatial information at deep features. By adding the position encoding to feature map $F_3$, the transformed features will become distinctive enough following position, which is crucial to the ability of feature matching.

Transformers and Attention extend to vision research, It made vision transformers networks (Ramachandran et al., 2019; Zhang et al., 2019) the latest trend in computer vision. Inspired by Chen et al. (2023), Sun et al. (2021), after feature extraction, we use the cross and self-cross attention Transformer modules, with the attention learning module to enhance feature representation by focusing on meaningful features. The attention is based on the vision transformer concept and uses convolution modules to extract Query, Key, and Values. The attention map is calculated according to Equation 1.

$$Attention(Q, K, V) = softmax(\frac{QK^T}{\sqrt{d_k}})V \qquad (1)$$

We apply feature map attention on both $F_2$ and $F_3$ scale feature maps. In each scale, the feature map performs cross-attention and self-attention corresponding to the feature map, for example, cross-attention is first applied to the $F_2^s$ and then self-attention is applied to itself. The same is true for $F_2^t$, $F_3^s$, and $F_3^t$ are applied attention in the same way.

## 3.4 CORRELATION MAP AND MARGINALIZATION

Hyper correlation concept was introduced in Min et al. (2021) to compute 4-D correlation across multi-scale feature maps for few-shot segmentation. We use a similar concept and introduce a module in our model that constructs a 4-D correlation map from cosine similarity after feature extraction. 4D correlation map between two feature maps $F_l^s, F_l^t$ for l=2,3 at pixel positions i,j and m,n are constructed using the cosine similarity as follows:

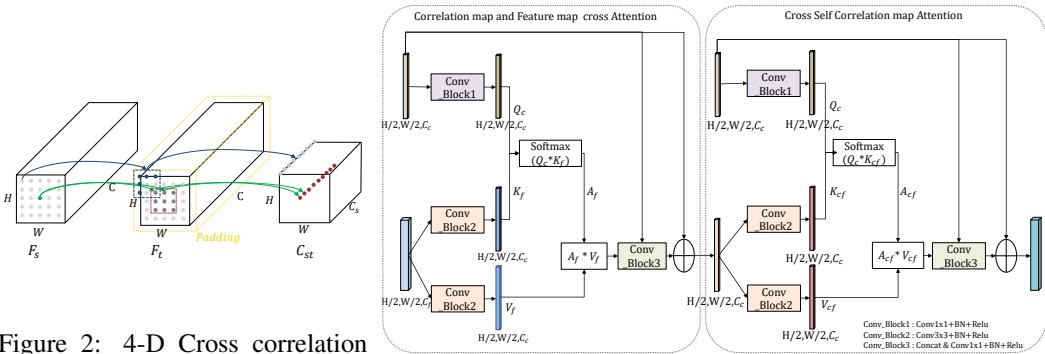

Figure 2: 4-D Cross correlation and marginalization.

Figure 3: Cross and Cross-Self Attention: Correlation map attention process using lower level feature map.

$$\boldsymbol{Cos}(\boldsymbol{F}^s(i,j), \boldsymbol{F}^t(m,n)) = \boldsymbol{ReLU}\left(\frac{\boldsymbol{F}^s(i,j) \cdot \boldsymbol{F}^t(m,n)}{||\boldsymbol{F}^s(i,j)|| \cdot ||\boldsymbol{F}^t(m,n)||}\right) \tag{2}$$

By assuming in text matching scenario the position of correspondence point within the neighboured of the queried point. Because in a comparison between text images, the target image may be rotated, blurred, or have white space or noise, but the same character is present in a similar position in both the source and the target. Instead of calculating a full dense 4-D correlation map, we just restrict the matching to neighboured points to the queried feature point. The neighboured is defined as $[-K_v, K_v]$ and $[-K_h, K_h]$ in both vertical and horizontal directions. We calculate the sparse correlation with a pre-defined range instead of the full correlation map which saves our computation time. Correlation map marginalization is the process of converting a 4-D correlation tensor into two 3-D correlation tensors.

As shown in Figure 2, we calculate the neighboured correlation map in the defined size of $(2 * K_v + 1)$, $(2 * K_h + 1)$ in each spatial pixel position of the feature map, and convert it into $C_c$ channel marginalized 3-D correlation tensor $C_{st}$ from feature maps $F_s$ to $F_t$. The process is the same for computing marginalized correlation from feature maps $F_t$ to $F_s$ by interchanging and the result is $C_{ts}$. The channel size of marginalized correlation map is $C_c = (2 * K_v + 1) * (2 * K_h + 1)$. The algorithm for computing $C_c$ channel marginalized correlation maps using cosine similarity is given in Algorithm 1 under Appendix A.1.

## 3.5 CORRELATION MAP CROSS AND CROSS-SELF ATTENTION

In addition to the feature map cross and cross-self attention and taking advantage of the attention in 3.3, we apply cross and cross-self attention mechanism on marginalized lower resolution correlation maps from the lower scales. Before applying the attention mechanism, the marginalized correlation maps $C_2^s$, $C_3^s$ and $C_2^t$, $C_3^t$ from the two levels are Deconvolution and up-sampled to the size of $(H/2, W/2)$ by shared Deconvolution and Up-sample layers. The up-sampled correlation maps $C_2^s$, $C_3^s$ and $C_2^t$, $C_3^t$ are each independently performed attention process with the top-level feature maps $F_1^s, F_1^t$. The attention process using transformer attention is shown in Figure 1, and 3. The $C_f$ channel feature map $F_1$ is used as the Key and Value for Attention. In order to match the channel size between $C_c$ channel query correlation map and encoded feature map $F_1$, the feature map is squeezed using Conv_Block2 generate new Key $K_f$ and Value $V_f$ and then attention is applied. Finally the input feature map and queried feature map $A_f * V_f$ are merged using Conv_Block3 and added to the input feature.

As a next step, cross-self attention is applied to the input correlation map using the following similar process as cross attention but the query and key features are from the cross-attention output as shown in Figure 3. The same process is applied on all 4 correlation feature maps using the respective image $F_1$ encoded feature.

### 3.6 SEGMENTATION MAP

Finally, the correlation feature maps from the attention module in each decoder module merged using the convolution process. The segmentation maps are generated using convolutions upsampling by a factor of 2 and applied Sigmoid activation at the end. The decoder part consists of 2 parallel decoder modules for each image $I_s$ and $I_t$. The output of the decoder $S_{st}, S_{ts}$ are two-way semantic segmentation change maps from $I_s$ to $I_t$ and from $I_t$ to $I_s$.

### 3.7 LOSS FUNCTION

$I_s$ and $I_t$ are the source image and target image respectively, the segmentation loss function is the combination of Dice Loss $L_d$ and Binary Cross-Entropy (BCE) loss $L_{bce}$ as given in equations as follows:

$$L_{st}(I_s, I_t) = L_d(S_{st}, G_{st}) + L_{bce}(S_{st}, G_{st}) \tag{3}$$

$$L_{ts}(I_t, I_s) = L_d(S_{ts}, G_{ts}) + L_{bce}(S_{ts}, G_{ts}) \tag{4}$$

where $S_{st}$ is s to t predicted segmentation map and $G_{st}$ is s to t ground truth segmentation map. For each pixel position $i$, $y_i$ and $p_i$ are ground truth and prediction at the pixel respectively, Dice Loss $L_d$ and Binary Cross-Entropy (BCE) loss $L_{bce}$ are given in equations as follows:

$$L_d = 1 - \frac{2 * y_i * p_i + \tau}{y_i + p_i + \tau}, \quad \tau = 1 \tag{5}$$

$$L_{bce} = -(y_i \log(p_i) + (1 - y_i) \log(1 - p_i)) \tag{6}$$

Our model has two segmentation maps so the total segmentation loss is average of both of them and the overall loss function is as follows:

$$L_{seg} = 0.5 * L_{st}(I_s, I_t) + 0.5 * L_{ts}(I_t, I_s) \tag{7}$$

### 3.8 TRAINING WITH SYNTHETIC DATA

Our TCD model inputs are the pair of unit text images and the ground truth is the changed area. Our model training data is generated by a synthetic image generator using a text corpus from English, Korean, Chinese, Numbers, and special characters. Each sample data consists of pair of images along with the ground truth change area. As we designed TCD as two-way segmentation, our ground truth consists of bidirectional segmentation maps. We assume the purpose of our model is to compare contract documents, it is typically compared between machine-readable documents and scanned documents. Therefore, the source image is generated using fixed background and the target image is generated using a random background by considering scan document image quality. In Figure 5 under Appendix A.2, we show some of the samples of our data generator.

From the samples, it can be observed that the segmentation map is a pseudo segmentation map of rectangular shape based on the width and height of the character in the image which is set to 1 if the particular position changed or 0. The data generator is designed in such a way that it can generate data with fixed heights and various widths of images to support dynamic widths of the unit sizes instead of fixing it. For batch training, create uniformly sized data by padding background pixels to the right side of text images in the training phase. Source and target images are generated by considering various real document changes while scanning, such as random text position, random character spacing, and random character positioning. Plus, to simulate various scanning and picturing quality distortions of images, we applied various augmentations such as bleed-through, blurring, noise, scaling, underline, over-line, cross-line, and rotations on the synthetic images. In each batch of data, we make sure the data is balanced so each batch consists of the same and changed data pairs. The same pair is generated using the same text corpus, while the change data pair is generated by randomly modifying some characters at random positions.

## 4 Experiments and Results

We validate the contribution of our model with several experiments. For benchmarking, we created unit image pair dataset in various different languages. In the benchmark results, we provide comparisons for each language, as well as for the entire evaluation dataset. We also prove the utility of each of the modules we proposed through ablation studies.

### 4.1 Test Dataset

As per the research literature, there are no open public datasets available on text image change detection for document comparison. So, we created new dataset for text image change detection. In the following subsections, we discuss in detail about different dataset prepared and used to study our model performance.

We prepared two kinds of datasets to evaluate the performance of segmentation and OCR separately. They are used, to benchmark with semantic segmentation, CD models, and OCR methods respectively. The segmentation test dataset is prepared from actually printed and scanned pair words from documents written in different languages such as English, Korean, Russian, and Chinese. The units are cropped at character using DUET (Jung et al., 2021) unit detector. These cropped units are merged randomly to make concatenated text images and created synthetic segmentation change ground truth similar way like training dataset. The same and diff pairs are generated using position in the full image and text ground truth. The dataset consists of totally 80k pairs, 10k same pair and 10k different pair from each of the 4 languages. The composition of the segmentation dataset is shown in Table 4 under Appendix A.3.

In addition, OCR test dataset is prepared from the public datasets DDI-100, LRDE-DBD plus our data created by printing and scanning the documents. DDI-100 consists of English, Russian, and Digits text, LRDE consists of French text, and our dataset is in Korean, and Chinese text. The dataset is prepared from paired images with the same content from the original and distorted or scanned using OCR ground truth. The text units are cropped from each document image using ground truth detection and text. Then, the dataset is grouped using OCR ground truth, and then the same and different pairs of datasets are created using text similarity. The different pair data are picked randomly that have a maximum of 4 character changes and a maximum length difference of 1. The composition of the OCR dataset is shown in Table 5 under Appendix A.3.

### 4.2 Train and Test methodology

We set the height of training data is fixed and set as 32. We use 3-scale feature maps and the channel sizes in each scale after FPN is size 64, 64, and 512. The neighboured value $K_v, K_h$ for correlation marginalization is set 2,4 and 1,2 for scale $H/4$ and $H/8$ features accordingly. Plus, segmentation loss value is scaled by a constant 10 during training. We train our model for 200 epochs from scratch with a batch size of 8. Finally, we evaluate our model performance on the test dataset.

To evaluate segmentation test dataset performance for benchmarking and ablation study, we use similar performance measures like other semantic segmentation and CD models such as precision, recall, F1 score, IoU, and overall accuracy. We include both the same and diff pair datasets in the evaluation dataset as the measures consider overall data pixel level performance. In addition, to evaluate OCR test data, we use image-level classification performance scores of the same and different pairs. By using the classification confusion matrix we calculate precision, recall, F1 score, and Accuracy for all the OCR benchmarking models.

### 4.3 Benchmark results

To check the performance capabilities of our model, we selected well-known or SoTA semantic segmentation and CD models such as U-Net (Ronneberger et al., 2015), SegFormer (Xie et al., 2021), BIT-CD (Chen et al., 2021), SARAS-Net (Chen et al., 2023), and OCR methods like Tesseract OCR (Smith, 2007), Multiplex OCR (Huang et al., 2021), PPOcr v3 (Li et al., 2022). In general, semantic segmentation and CD models are not trained on text datasets, for a fair comparison, we retrain them using our synthetic training data generator similar way to our model for 200 epochs each with default settings. We evaluate all the models with metrics discussed in the test methodology

Table 1: Quantitative average results of segmentation benchmark.

| Model | Pre. | Rec. | IoU | F1. | OA. |
|-------|------|------|-----|-----|-----|
| U-Net | **64.0** | 65.7 | 64.6 | 47.8 | 84.8 |
| SegFormer-B5 | 60.9 | 81.0 | 69.5 | 53.2 | 84.9 |
| BIT-CD | 63.9 | 72.1 | 67.6 | 51.2 | 85.4 |
| SARAS-Net | 62.1 | 70.5 | 65.9 | 49.2 | 84.6 |
| TCD (ours) | 63.7 | **81.4** | **71.5** | **55.6** | **86.3** |

Table 2: Quantitative average results of OCR benchmark.

| Model | Pre. | Rec. | F1. | Acc. |
|-------|------|------|-----|------|
| Tesseract OCR | 90.1 | **99.9** | 93.2 | 88.7 |
| Multiplex OCR | 70.7 | 96.6 | 80.7 | 71.5 |
| PPOcr v3 | 95.6 | 99.8 | 97.5 | 97.1 |
| TCD (ours) | **99.5** | 99.2 | **99.4** | **99.2** |

Figure 4: Qualitative results of different segmentation models on paired source and target input unit images the. Output segmentation of our model is the maximum of two way segmentation maps.

subsection. In order to evaluate OCR models, we used available public pre-trained models for each language.

Table 1 illustrates performance comparison with SoTA segmentation and CD methods on 4 language segmentation datasets, respectively. Clearly, our model outperforms all SoTA methods on average on language datasets with 2 points in F1 score and IOU and 1 point in accuracy. And, also TCD maintains similar performance across all 4 language datasets as shown in Table 6 under Appendix A.3. To visualize the prediction results, the results of different methods on the above 4 language datasets are shown in Figure 4. The top rows show all the changed text scenarios and the bottom rows show the same text scenarios. In the figure white color is changed text area and the black color is the same text area. Even though our model generates two segmentation maps, here we just show the maximum result of two outputs to compare with benchmark single output models. The actual source and target pair inputs are not the same size, so before inference of each model result, the input's height is resized and width right padded to make them the same size. From the prediction results, it can be observed that TCD generates a segmentation map sharper, less noisy, and close to the ground truth for different cases of text changes and text change positions, other methods fail in some cases to detect real changes. For the same text image input pairs, all other methods generate noisy and false positive change maps.

Similarly, Table 2 illustrates performance comparison with SoTA OCR on of 5 languages and digits average. We classify text pair images as either the same or changed. Our model output two way segmentation maps are merged into one map and then the input text image pair classified as same if the number of change pixels all are 0 otherwise different. From Table 7 under Appendix A.3 OCR results tend to have a dependency on language. In particular, most methods results are good including Digits and English, but in some cases like Russian, the performance is slightly worse. Most of the OCRs methods still show good results, but for each language OCR needs its own weight. Although we have not mentioned it in detail here, the OCR recognition itself is often wrong, because

Table 3: Ablation study results to show the effect of each module (all language average) on segmentation dataset.

| Model/Module | CM | FA | CA | TW | Pre. | Rec. | F1 | IoU | OA |
|---|---|---|---|---|---|---|---|---|---|
| TCD v1 | X | X | X | ✓ | 62.8 | 70.4 | 66.3 | 49.7 | 84.8 |
| TCD v2 | ✓ | X | X | ✓ | 62.2 | 80.3 | 70.1 | 54.0 | 85.4 |
| TCD v3 | ✓ | ✓ | X | ✓ | 62.7 | 76.9 | 69.1 | 52.8 | 85.4 |
| TCD v4 | ✓ | X | ✓ | ✓ | **65.1** | 75.7 | 69.9 | 53.8 | **86.3** |
| TCD v5 | ✓ | ✓ | ✓ | X | 64.9 | 76.9 | 70.3 | 54.3 | **86.3** |
| Ours | ✓ | ✓ | ✓ | ✓ | 63.7 | **81.4** | **71.5** | **55.6** | **86.3** |

of the reason diff pair recall is relatively high, but the precision of predicting the same pair as same is relatively low. The multiplex OCR uses only one weight, but the performance varies greatly depending on the language. On the other hand, our model shows high performance regardless of language.

## 4.4 ABLATION STUDY

We performed an ablation study of different modules in our TCD model using a segmentation benchmark dataset. The modules we studied are correlation and marginalization (CM), encoder feature map attention (FA), decoder correlation map attention (CA), and one-way (OW) vs two-way (TW) segmentation maps. We also compared with a model by removing all the modules specified above and by replacing the CM module with Conv layers which is a basic model. So we study the performance of each module by adding or removing it in a progressive manner.

**Correlation and marginalization (CM)** To study the effect of the CM module, we first trained TCD v1 by replacing CM with basic Conv layers and TCD v1, TCD v2 are trained for 200 epochs each. Then they are evaluated on the segmentation dataset, from Table 3 it can be observed that by adding CM module recall, F1, IoU, and OA are improved by 10, 4, 6, and 1 percent respectively. The ablation study shows that a correlation map of features can extract good meaningful information for change detection instead of using a simple feature map.

**Cross and cross/self-attention (FA or CA)** Feature cross and cross self-attention modules (FA) are applied across the source and target to improve the feature map enhancing the features useful for the change detection and scale changes between two images. Correlation cross and cross self-attention (CA) modules are applied to the lower resolution correlation map to improve the change segmentation map. From Table 3 TCD v3, TCD v4 rows, it can be observed that they have an effect on precision and overall accuracy even though there is degradation in recall.

**Two-Way segmentation train (TW)** We trained TCD v5 with only a one-way segmentation map like other semantic segmentation models by taking the maximum of two-way model output, training with single segmentation map loss instead of TCD two way loss. By Comparing the performance between TCD v5 and ours from Table 3, Our model precision decreased slightly but other metrics are improved. The main purpose of document comparison is to find modified words or areas. Our model two way segmentation result is useful to analyze not only simple changes but also how they changed from each others point of view.

## 5 CONCLUSION

In this paper, we propose the TCD model which is vital block for imaged document comparison in various languages. Our model is language-independent because we adopt image comparison instead of text recognition. TCD model architecture is designed with multi-scale features and correlation marginalized maps for text image change detection at a small unit level of the document. Proposed model is robust to various changes in the text unit image and doesn't require any preprocessing such as text alignment or scale alignment. In addition, we use a correlation map with feature map, cross, and cross-self transformer-based attention for improved change segmentation. From the experimental results, we show that our model is well generalized for multilingual documents irrespective of the language. And also our model works well for other language documents which are not in training text corpus. From the benchmarking results, our model outperformed other semantic segmentation models on average by a good margin and performed similar to OCR methods that use a model for each language.

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

# A    APPENDIX

## A.1    CORRELATION MAP AND MARGINALIZATION

---

**Algorithm 1** Correlation and Marginalized Correlation map

---

1: **Input**: Assume $F_l^s, F_l^t$ source and target feature maps at same scale for l=2,3
2: Normalize source and target feature maps along the channel dimension
3: Pad source and target feature maps by $K_h, K_v$ in horizontal and vertical direction on both sides results in $F_l^{s'}, F_l^{t'}$
4: Initialize a empty correlation map $C_{st}, C_{ts}$ of size $C_c$x$H_l$x$W_l$
5: Let $K_w = (2 * K_h + 1), K_h = (2 * K_v + 1)$
6: Let $i = 1, j = 1$
7: **while** $i <= K_w$ **do**
8:     **while** $h <= K_h$ **do**
9:         $t1 = (F_l^{t'}[i : i + H, j : j + W] * F_l^s)$
10:         $t2 = (F_l^{s'}[i : i + H, j : j + W] * F_l^t)$
11:         $C_{st}[i * K_w + j, :, :]=\Sigma_c t1$
12:         $C_{ts}[i * K_w + j, :, :]=\Sigma_c t2$
13:     **end while**
14: **end while**
15: **Output**: Marginalized feature correlation map $C_{st}, C_{ts}$ for each multi-scale feature map set $F_l^s, F_l^t$

---

## A.2    SYNTHETIC TRAIN AND VALID DATASET

We made a corpus dataset of a total of 10000 words that consist of English, Korean, Chinese, Numbers, and special characters. Training data source image is made from a text corpus and modified characters are picked randomly from other corpus text. All data is created as a pair of source images and a matching target change map image. In each batch of images, the same and change pairs are equally generated for data balancing. Similarly, we used another 5000 text corpus data for generating validation dataset, which is used for selecting the best model. The best model is selected based on validation Intersection Over Union (IOU) of the segmentation. The sample data is shown in Figure 5.

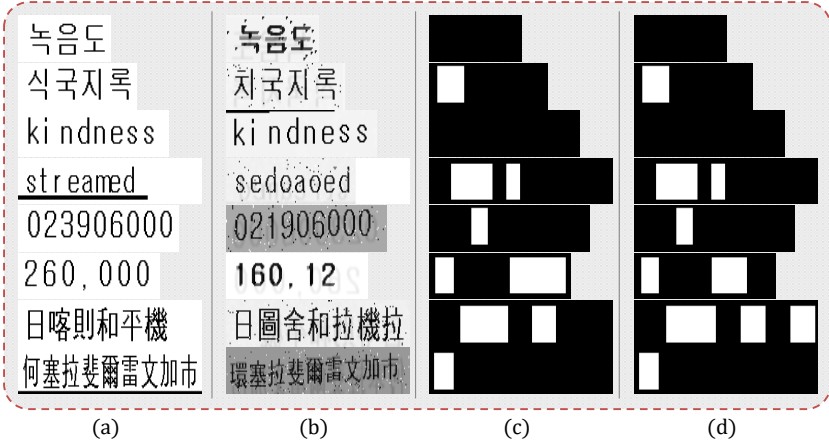

Figure 5: Sample results of synthetic training data : (a) source, (b) target, (c) segmentation ground truth from source to target, (d) segmentation ground truth from target to source.

Table 4: Segmentation dataset taxonomy, where Same is same pair and Diff is different (change) text image pair.

| Language | English | Russian | Korean | Chinese |
|---|---|---|---|---|
| Same | 10000 | 10000 | 10000 | 10000 |
| Diff | 10000 | 10000 | 10000 | 10000 |
| Total | 20000 | 20000 | 20000 | 20000 |

Table 5: OCR dataset taxonomy, where Same is same pair and Diff is different (change) text image pair.

| Language | English | Russian | French | Korean | Chinese | Digits |
|---|---|---|---|---|---|---|
| Same | 50002 | 47462 | 19234 | 49998 | 17538 | 11167 |
| Diff | 49998 | 52538 | 30766 | 39772 | 32462 | 38833 |
| Total | 100000 | 100000 | 50000 | 50000 | 50000 | 50000 |

## A.3 BENCHMARK DATASET AND RESULTS

Table 6 and Table 7 show the different benchmark models detailed performance across different language datasets for both segmentation and OCR datasets. Each metric is evaluated on each language data individually and then averaged across the languages.

Table 6: Quantitative results of segmentation benchmark for different languages and its average.

| Language | English | | | | | Korean | | | | | Chinese | | | | |
|---|---|---|---|---|---|---|---|---|---|---|---|---|---|---|---|
| Model | Pre. | Rec. | F1 | IoU | OA | Pre. | Rec. | F1 | IoU | OA | Pre. | Rec. | F1 | IoU | OA |
| U-Net (Ronneberger et al., 2015) | 65.8 | 54.5 | 59.6 | 42.5 | 83.2 | 63.1 | 71.8 | 67.2 | 50.6 | 85.6 | 63.5 | 67.8 | 65.5 | 48.7 | 84.5 |
| SegFormer-B5 (Xie et al., 2021) | 61.2 | 82.7 | 70.3 | 54.2 | 84.1 | 61.1 | 83.5 | 70.6 | 54.5 | 85.7 | 62.1 | 78.0 | 69.2 | 52.9 | 84.9 |
| BIT-CD (Chen et al., 2021) | 62.8 | 77.5 | 69.4 | 53.1 | 84.4 | 64.4 | 73.9 | 68.8 | 52.4 | 86.3 | 62.9 | 61.1 | 62.0 | 44.9 | 83.8 |
| SARAS-Net (Chen et al., 2023) | 62.5 | 79.7 | 70.0 | 53.9 | 84.5 | 62.3 | 70.4 | 66.1 | 49.3 | 85.2 | 62.0 | 65.9 | 63.9 | 46.9 | 83.8 |
| TCD(ours) | 64.1 | 81.3 | 71.7 | 55.9 | 85.4 | 63.2 | 84.2 | 72.2 | 56.5 | 86.7 | 63.3 | 76.5 | 69.3 | 53.0 | 85.3 |

| Language | Russian | | | | | Avg. | | | | | | | | | |
|---|---|---|---|---|---|---|---|---|---|---|---|---|---|---|---|
| Model | Pre. | Rec. | F1 | IoU | OA | Pre. | Rec. | F1 | IoU | OA | | | | | |
| U-Net (Ronneberger et al., 2015) | 63.6 | 68.7 | 66.0 | 49.3 | 85.7 | **64.0** | 65.7 | 64.6 | 47.8 | 84.8 | | | | | |
| SegFormer-B5 (Xie et al., 2021) | 59.1 | 79.7 | 67.9 | 51.4 | 84.8 | 60.9 | 81.0 | 69.5 | 53.2 | 84.9 | | | | | |
| BIT-CD (Chen et al., 2021) | 65.6 | 75.8 | 70.3 | 54.3 | 87.1 | 63.9 | 72.1 | 67.6 | 51.2 | 85.4 | | | | | |
| SARAS-Net (Chen et al., 2023) | 61.8 | 66.0 | 63.8 | 46.8 | 84.9 | 62.1 | 70.5 | 65.9 | 49.2 | 84.6 | | | | | |
| TCD (ours) | 64.2 | 83.6 | 72.6 | 57.0 | 87.3 | 63.7 | **81.4** | **71.5** | **55.6** | **86.3** | | | | | |

Table 7: Quantitative results of OCR benchmark for different languages and its average.

| Language | English | | | | Korean | | | | Chinese | | | | Russian | | | |
|---|---|---|---|---|---|---|---|---|---|---|---|---|---|---|---|---|
| Method | Pre. | Rec. | F1 | Acc | Pre. | Rec. | F1 | Acc | Pre. | Rec. | F1 | Acc | Pre. | Rec. | F1 | Acc |
| Tesseract OCR (Smith, 2007) | 87.7 | 99.9 | 93.4 | 92.9 | 97.8 | 100.0 | 98.9 | 98.2 | 84.2 | 100.0 | 91.4 | 89.7 | 89.7 | 99.8 | 94.5 | 93.9 |
| Multiplex OCR (Huang et al., 2021) | 56.2 | 98.3 | 71.5 | 60.9 | 84.6 | 95.7 | 89.8 | 82.7 | 75.4 | 87.2 | 80.9 | 73.2 | 55.6 | 98.0 | 70.9 | 57.8 |
| PPOcr v3 (Li et al., 2022) | 99.3 | 100.0 | 99.6 | 99.6 | 99.5 | 99.9 | 99.7 | 99.5 | 99.0 | 100.0 | 99.5 | 99.3 | 79.8 | 99.4 | 88.5 | 86.5 |
| TCD (ours) | 99.4 | 99.5 | 99.4 | 99.4 | 100.0 | 99.4 | 99.7 | 99.5 | 99.7 | 99.9 | 99.8 | 99.7 | 99.4 | 99.4 | 99.4 | 99.4 |

| Language | French | | | | Digits | | | | Avg. | | | | | | | |
|---|---|---|---|---|---|---|---|---|---|---|---|---|---|---|---|---|
| Method | Pre. | Rec. | F1 | Acc | Pre. | Rec. | F1 | Acc | Pre. | Rec. | F1 | Acc | | | | |
| Tesseract OCR (Smith, 2007) | 94.2 | 99.7 | 96.8 | 96.0 | 87.3 | 99.9 | 93.2 | 88.7 | 90.1 | **99.9** | 93.2 | 88.7 | | | | |
| Multiplex OCR (Huang et al., 2021) | 65.5 | 98.5 | 78.7 | 67.1 | 87.1 | 98.3 | 92.3 | 87.3 | 70.7 | 96.6 | 80.7 | 71.5 | | | | |
| PPOcr v3 (Li et al., 2022) | 96.4 | 99.6 | 98.0 | 97.5 | 99.8 | 100.0 | 99.9 | 99.9 | 95.6 | 99.8 | 97.5 | 97.1 | | | | |
| TCD (ours) | 98.9 | 97.4 | 98.1 | 97.7 | 99.7 | 99.8 | 99.7 | 99.6 | **99.5** | 99.2 | **99.4** | **99.2** | | | | |

