# OpenReview forum: "TCD: TEXT IMAGE CHANGE DETECTION FOR MULTILINGUAL DOCUMENT COMPARISON"
_ICLR.cc/2024/Conference — Submitted to ICLR 2024_

### Official Review · Reviewer_RtiC · 2023-10-31

**Soundness:** 3 good
**Presentation:** 3 good
**Contribution:** 2 fair
**Rating:** 5
**Confidence:** 3

**Summary:**

This paper focuses on multilingual document comparison. OCR-based methods highly rely on the performance of recognition, restricting the potential toward untrained languages. Hence, the authors propose an image-based method called TCD to get rid of OCR process. TCD only compares the embedding of source and target image bidirectionally to detect inconsistency where the correlation marginalization process is the key. A new text image change detection test dataset is presented. Experiments demonstrate the effectiveness of different modules in TCD and the full setting achieves the SoTA performance.

**Strengths:**

1.	The structure of the article is well organized.
2.	The proposed Correlation Marginalization sounds good to find inconsistency. And the two-way segmentation approach is a targeted design and effectively improves the performance. The proposed method is independent to OCR tools and can be extended to unseen multilingual data.
3.	A data synthesis approach for multilingual document is proposed. A new test dataset for text image change detection is created.
4.	The proposed method achieves the SoTA overall performance on segmentation and OCR benchmark. The experiments on Russian texts shows the potential of TCD to handle untrained language.

**Weaknesses:**

1.	I wonder how the source image is obtained in actual situations. The downstream applications of should be highlighted.
2.	The analysis of experiments needs to be enriched. For example, why the precision of U-Net is better among the benchmark methods?
3.	The Correlation Marginalization is designed to save computation time. The ablation study only compares with Conv layer while ignoring the straightforward comparison to 4-D Correlation Map.
4.	It would be better to provide some cases compared to OCR methods. For example, the false recognition leads to poor performance.
5.	The writing needs to be improved. For example, in Sec 3.1 line 8, the conjunction ‘However’ is confusing. I kindly suggest the authors to polish the language.

**Questions:**

In the second column of Fig 4, the character ‘s’ is annotated as changed, but it is aligned with the ‘s’ in source image. What’s the criteria of annotating such cases?

---

### Official Review · Reviewer_6nRM · 2023-11-01

**Soundness:** 2 fair
**Presentation:** 1 poor
**Contribution:** 1 poor
**Rating:** 3
**Confidence:** 5

**Summary:**

The paper proposes a new framework and a data synthesis method for a new research topic: Image Document Comparison. This topic, according to the explanation in the paper, aims to detect changed text locations such as changed, added, or deleted in document text images at character level.
The framework is built based on an Encoder-Decoder architecture, which consists of a share-weight encoder, a correlation marginalization module and a parallel decoder, for predicting the segmentation map of different characters between the source text image and target text image. And the image pair is collected and synthesized by the proposed data synthesis method, which includes using a synthetic image generator for training data and cropping-and-sticking character image manually from real world document image for testing data.
According to the authors, this paper first proposed a linguistic-free model with a unique correlation marginalization process for text image comparison. Besides, they present a new text image change detection test dataset.

**Strengths:**

With respect to originality, the paper put forward a new research topic of text image change detection. This topic seems to be similar to another emerging topic in text-image research area, named tamper detection, for both predict the segmentation map between paired images, while the former focuses on the semantic difference at linguistic level and the later aims to detect the tamper area at visual level. The proposed topic appears to be novelty for seeking a linguistic-free method for text image comparison, but lack of necessity and practicability in reality.
With respect to quality, the paper adopts a complex and manually-designed encoder-decoder architecture which mainly concludes FPN and attention block. It is lack of interpretability for the necessity and effectiveness of such complicated design and inconvincible for the robustness.
As to the clarity, the paper is of poor readability with a lot of grammatical errors.
With respect to significance, there is hardly any word about the application or reality necessity of the proposed “Image Document Comparison”, nor any example from reality is shown in the paper since both the training and testing dataset are synthesized or manually concatenated by authors.

**Weaknesses:**

1.	The amount of the grammatical mistakes in the paper is unbearable.
2.	There isn’t enough word about the application or necessity of text image comparison through the passage, which makes it doubtful for the necessity of the research.
3.	The works reviewed in Section “Related Work” is quite different from the topic proposed in this paper. It will be better to discuss about and compare with the works relevant to universal image comparison if this paper is the first work in text image comparison.
4.	According to Section 3, a component called “cross-self attention” is frequently used in several module in proposed framework. The authors claim that it benefits the ability of feature matching between feature maps. This naturally aroused a question: why a common transformer block is not used. The paper is lack of experiment for proving the effectiveness of proposed component.

**Questions:**

1.	Where does the necessity of text image comparison task lie in?
2.	Can the proposed dataset well represent the scenarios that will actually be encountered?

---

### Official Review · Reviewer_qbQL · 2023-11-01

**Soundness:** 2 fair
**Presentation:** 2 fair
**Contribution:** 2 fair
**Rating:** 5
**Confidence:** 4

**Summary:**

This article presents a new method for detecting changes in document images. Unlike conventional OCR-based methods, this method directly compares thumbnail images of lines of text to identify areas of difference. The method is based on a deep neural network combining feature extraction with a ResNet and multi-scale attention modules. The proposed method is compared to image-based methods and different OCRs on a dataset newly created by the authors. This comparison shows that the proposed method outperforms image-based methods and gives performances similar to those obtained with OCR.

**Strengths:**

- an original change detection method, precisely described
- a comparison with semantic segmentation, CD and OCR methods
- an ablation study
- an honest analysis of the results

**Weaknesses:**

## Why  Multilingual documents ?

The need for a specific method for analysing multi-lingual documents is not really argued.

## The references in the introduction and related work section are mostly irrelevant.

 Very few references to previous work in document change detection are given. In particular :

First paragraph:
- 3 references to the history of OCR are not needed; TAUSCHECK is badly formatted.
- idem, 3 general references to DL are not needed
- DL OCR references are a bit outdated, most recent is 2016, 7 years old
- give a reference of a transformer applied to document processing, not a generic transformer

- I don't understand why Taafti2016 is relevant for image comparison.

At the end of the introduction, the context is not clear: are we talking about forgery, detecting changes in different versions of a document that should be identical? are we talking about scanned documents or image PDFs?

Related work: This section should focus on document comparison methods, not on OCR subsystems such as text detection and recognition and generic semantic segmentation of images. The references to text detection and text recognition methods cannot be exhaustive, and the reader does not know whether all these methods have been used in the context of document comparison.

Missing references;
- Rajiv Jain and David Doermann. VisualDiff: Document Image Verifica- tion and Change Detection. In 2013 12th International Conference on Document Analysis and Recognition, pages 40–44, August 2013. ISSN: 2379-2140.
- Rajiv Jain and David Doermann. Localized document image change detection. In 2015 13th International Conference on Document Analysis and Recognition (ICDAR), pages 786–790, August 2015.
- Noo-ri Kim, YunSeok Choi, HyunSoo Lee, Jae-Young Choi, Suntae Kim, Jeong-Ah Kim, Youngwha Cho, and Jee-Hyong Lee. Detection of document modification based on deep neural networks. Journal of Ambient Intelligence and Humanized Computing, 9(4):1089–1096, August 2018.
- Comparison of scanned administrative document images, 2020, https://arxiv.org/abs/2001.10785

## The method is not applicable to a full document

It is not clear how the method is applied to a complete document. It seems that lines of text need to be detected first, as the method is applied to lines of text. One limitation that seems very important is that you also have to compare the lines two by two, but how do you do this on an entire document? What if the documents don't have the same number of lines? What if the text is offset and overflows onto the next line? All the lines will be detected as different just because of an offset.

## Training and testing with synthetic data only, on a single database, produced by the authors.

All the data used appears to be synthetic. There may be no datasets with actual documents for detecting changes in documents, but it would be useful to be more precise on this point.

## No open-source code and the database is not distributed.

**Questions:**

- why are there two independent decoders? why not only one output? It seems that the 2 outputs are identical.
- regarding the data, the 2 categories, same and diff, are balanced in training and testing. Is it realistic? We expect the diff sequences to be more rare than the identical sequences; is this experimental setup realistic?
- the data seems to consist of lines of text of just a few letters or words, according to the examples presented. What are the statistics for the data in terms of size (pixel) and number of characters? It is realistic ?
- Table 6: same question with the metric: if the modifications are rare, precision/recall/F1 are not suitable for an evaluation, unless only the modified pixels are considered in the metric.
- Table 7: PPOcr V3 is as good as TCD, except for Russian, so that the average is in favour of TCD. But for all the other languages, there is no advantage in using TCD over a standard OCR. This very important result is only presented in the appendix, which is misleading.
- how did you train the semantic segmentation models with 2 inputs and 2 outputs ?

---

### Official Review · Reviewer_ygsh · 2023-11-06

**Soundness:** 1 poor
**Presentation:** 1 poor
**Contribution:** 2 fair
**Rating:** 3
**Confidence:** 5

**Summary:**

The paper presents a technique to detect changes in multi-lingual documents without performing OCR explicitly. It utilizes a unit-level text image-to-image comparison by building a correlation marginalization process on each feature surroundings to detect text-change between source and target documents.

The proposed network uses an encoder-decoder model. In the encoder, a Siamese network takes in source and target images. Using ResNet as an FPN backbone network, it produces 3 mutli-scale feature pyramid map pairs (of sizes N/2, N/4, N/8). Using ideas from transformers, positional encoding is applied on N/8 feature vector, followed by cross attention and self-cross attention (source and target images). Assuming changes are local, a cross-correlation and marginalization map is constructed from cosine similarity. In the decoder, cross attention and self-cross attention is applied independently on source and target marginalized correlation maps. Dice and binary cross-entropy loss are used as loss functions.

Training is done with synthetic data. Evaluation is performed across various scripts like English, Korean Russian and Chinese. For segmentation maps, the proposed TCD approach outperforms the others in F1 and IoU scores. TCD also gets the best OCR average F1 results.

**Strengths:**

The primary contribution of the paper is the proposed Change Detection Model Architecture that combines the best of Transformer model and Siamese network to compare changes across two documents. Assuming the changes are local, the addition of correlation marginalization module on cross-attention and self-cross attention feature maps help introduce a similarity concept.

**Weaknesses:**

There are several weaknesses of the paper:
1. Clarity in writing: It is highly recommended to get the paper edited through a native English speaker. The paper gets progressively harder to read. More importantly, Section 4, arguably one of the most important sections, is not coherent due to lack in sentence formulation. E.g. the sentences like below are extremely hard to make sense of:
- We set the height of training data is fixed and set as 32.
- Our model output two way segmentation maps are merged into one map and then the input text image pair classified as same if the number of change pixels all are 0 otherwise different.
- With vigorous research in document analysis (Huang et al., 2022; Fang et al., 2021), this served as a momentum for document comparison technology to be released as off-the-shelf software in the field of research
- it can be observed that the segmentation map is a pseudo segmentation map of rectangular shape based on the width and height of the character in the image which is set to 1 if the particular position changed or 0.

Apart from these, characters are incorrectly capitalized, used of 'however' and 'in general' is excessive and incorrect

2. Scientific vocabulary: The paper lacks scientific vocabulary. E.g. Abstract states that the proposed model outperforms 'by a good margin'. It is always recommended to be quantitative in such claims. 'Input' and 'Queried' vocabulary is mixed with 'source' and 'target' vocabulary in section 3.5. Section 4.4 states accuracy improvements in whole/integer numbers. Please be specific in how much improvements there were.

3. Section 4 is the weakest. Please provide clear instructions on how the data was generated. There aren't any details on OCR model and how OCR is performed across a source and target images, which fundamentally differ from each other. Lastly, the ablation study does not stress on the need for such a complex system. The improvements are minimal.

**Questions:**

1. How was data generated? Section 4.1 states that 'merged randomly to make concatenated text images'. Does this mean there are random words that don't make sense?

2. Are there only 80k image pairs across 4 languages? They seem too low to train and test a transformer based model. How was training done exactly? Was it per language basis?

3. Was every SOTA model stopped after 200 epochs or they were stopped right before overfitting?

4. No details are provided on how OCR classification is done! Does this mean OCR is applied to only that subset where image pair is classified as same?

 Once the paper is rewritten, perhaps more questions would emerge due to a better understanding of the paper.

---

### Meta-Review · Area_Chair_WAZk · 2023-12-06

**Metareview:**

This paper on OCR-less comparison of documents with transformers has received 4 critical reviews, which raised issues on

- Clarity in presentation and writing,
- Lack of details,
- Lack of positioning with respect to prior work,
- The method being not applicable to a full document,
- Issues with the experimental setup, the method being tested on synthetic data only,
- Missing baselines,
- Lukewarm improvements of the key design choices evaluated in the ablation studies.

The authors did not provide a rebuttal, so there was nothing to discuss.
The AC judges that the paper is not suitable for publication at ICLR 2024.

**Justification For Why Not Higher Score:**

-

**Justification For Why Not Lower Score:**

-

---

### Decision · Program_Chairs · 2024-01-16

Reject